# Improving Polyp Classification in Colonoscopy using Self-Supervised Learning with Side Information

1 Machine learning and deep learning are rapidly transforming medical image analysis, offering promis-
2 ing avenues to improve diagnostic accuracy and efficiency across numerous clinical applications.
3 Among the applications that can benefit significantly from these advances is the detection of colorectal
4 cancer (CRC), a major global health concern with approximately two million new cases detected
5 annually [12]. Most CRCs originate from adenomatous polyps, whereas hyperplastic polyps pose
6 limited risk of transitioning to cancer [5], making the classification of polyps into these subgroups
7 important to ensure correct treatment. Despite the importance of colonoscopy, it remains highly
8 operator-dependent, and variations in visual perception and clinical skill reduce the effectiveness of
9 screenings [8]. AI-based systems have been proposed to assist in polyp detection and classification,
10 but they typically rely on large-scale labelled datasets — which are costly and time-consuming to
11 obtain. Self-supervised learning (SSL) offers a promising alternative by enabling models to learn
12 useful representations from unlabeled data. Some of the most successful SSL approaches are joint
13 embedding architectures (JEAs), which align representations of augmented views of the same input.
14 These methods are motivated by the *MultiView assumption* [15]: the relevant information is shared
15 across augmented views, and aligning these views encourages the encoder to learn useful representa-
16 tions. Modern JEAs, such as SimCLR, Barlow Twins, and Masked Siamese Networks, have achieved
17 outstanding results relying on this assumption [6, 3, 1, 2, 10, 16, 9]

18 However, the MultiView assumption can be overly permissive. It does not distinguish between
19 task-relevant and task-irrelevant (nuisance) information that may be shared across views. In settings
20 such as colonoscopy, augmented views often preserve the strong background textures, irrelevant to
21 downstream diagnostic tasks. Standard SSL methods may entangle such nuisance features with the
22 more subtle task-relevant signals, degrading downstream performance. To address this, we introduce
23 the *Nuisance-Free MultiView* (NF-MV) assumption, an information-theoretic perspective on the
24 MultiView setting that explicitly excludes shared nuisance structure from the representation. Under
25 NF-MV, we frame the goal of SSL as learning representations sufficient for the task while being
26 invariant to nuisance information. We implement this framework using *side information*—auxiliary
27 data that shares nuisance structure but lacks task-relevant information—and penalize representational
28 overlap using a Jensen-Shannon divergence between main and side representations. This leads to
29 a simple and general extension of standard joint embedding objectives, suited to the difficulties of
30 representation learning in medical domains such as colonoscopy. Experiments show that our method
31 attains polyp-classification performance comparable to that of models pre-trained on substantially
32 larger private datasets.

**Applications in Medical Imaging and Endoscopy.**

34 SSL is set to become a key tool in medical and endoscopic image analysis. For instance, Wang et al.
35 [16] aligns spatiotemporal views to train encoders on endoscopy videos. Hirsch et al. [9] applied
36 the Masked Siamese Network approach to endoscopic video analysis, while $M^2CRL$ [10] combines
37 contrastive learning and masked image modelling, achieving impressive results. These methods
38 typically rely either on private datasets or curated clips that emphasise frames with visible polyps.
39 For example, $M^2CRL$ leverages 10 publicly available datasets totalling over 33,000 videos and 5.5

million frames, but primarily focuses on sequences where non-polyp frames have been filtered out. In contrast, full-length colonoscopy videos are dominated by *negative* frames. The REAL-Colon dataset [4], which we use for pre-training in our colonoscopy experiments, reflects this distribution: 87.6% of frames contain no polyps. Developing methods and frameworks that can effectively utilise this under-explored redundancy in real-world datasets has been a central motivation for our work.

**NF-MV Assumption**

In the standard MultiView SSL setting for JEAs we assume access to one unlabeled dataset $\mathcal{X}$, and some stochastic augmentation $A$. We define the set of paired views $x_1, x_2 \sim A(x)$. By the MultiView assumption, the downstream tasks optimized during pre-training can be expressed as:

$$\mathcal{T} = \{y \ : \ I(y; x_2 | x_1) < \epsilon, \ I(y; x_1 | x_2) < \epsilon\}, \quad \epsilon > 0. \tag{1}$$

In realistic settings, shared but irrelevant factors often persist across augmentations and become entangled with the learned representation. Based on this, we propose a new perspective on the MultiView assumption: by defining what to consider as a nuisance, it is possible to control what the algorithm considers as relevant or irrelevant information. That is, the modeller specifies a structure $n$ that should be considered irrelevant, and this nuisance specification induces a family of tasks for which the nuisance carries no label information.

**Assumption 1** (Nuisance-Free MultiView Assumption (NF-MV)). *Let $x_1, x_2$ be two views of an input $x$, and let $n_1, n_2$ be nuisance variables extracted from $x_1, x_2$, respectively. We assume:*

$$I(y; x_2 \mid x_1) \leq \varepsilon, \quad I(y; x_1 \mid x_2) \leq \varepsilon, \quad and \quad I(y; n_1) = I(y; n_2) = 0$$

*Then we say the Nuisance-Free MultiView assumption holds for y.*

If we substitute the MultiView assumption for the proposed Nuisance-Free MultiView Assumption, a new, strictly smaller, set of tasks arise.

**Definition 1** (NF-MV Induced Task Set). *Given nuisance $n$, we define the set of induced tasks as:*

$$\mathcal{T}_{nf}(n) := \{y \ : \ I(y; x_2 \mid x_1) \leq \varepsilon, \quad I(y; x_1 \mid x_2) \leq \varepsilon, \quad I(y; n) = 0\}$$

This task set consists of all labels that can be predicted equally well from either view *and* are independent of the nuisance. Once the modeller specifies a nuisance variable $n$, this isolates the subset of MultiView-induced tasks that are consistent with the modelling choice of what information should be ignored. If $n$ is sufficiently well-defined, then $\mathcal{T}_{nf}(n)$ captures the tasks which we are interested in, allowing us to target the learning without access to fine-grained labels.

**Leveraging Side Information via Jensen-Shannon Divergence**

As motivated by the analysis above, it is preferred to learn an encoder that disentangles the nuisance features from relevant ones. To pinpoint nuisance structures we assume access to side information $\mathcal{S}$, that contains information that is (approximately) irrelevant but overlapping with the main dataset $\mathcal{X}$. The nuisance is then defined as the structural overlap between $\mathcal{X}$ and $\mathcal{S}$. When working with joint embedding models in a single feature space, there are additional subtleties to consider. First, we need to have informative representations of the side information $s \sim \mathcal{S}$ in order to disregard it. If the representations $f_\theta(s)$ are unreliable, it is not possible to disentangle the representations of the main data $f_\theta(x)$ between relevant and irrelevant structures. Second, estimating and controlling mutual information in the extremely high-dimensional feature spaces where JEA methods operate is notoriously difficult. Estimators such as CLUB [7] and L1Out [13] suffer from high variance and bias in these high-dimensional spaces. Moreover, since they require neural network parametrization, the training procedure becomes more complex. Taking these considerations into account, we propose a simple objective for using side information with JEAs. Let $z = f_\theta(A(\omega))$, where $\omega \sim M_\alpha = \alpha \mathcal{X} + (1 - \alpha)\mathcal{S}$, and let $B_\alpha \in \{0, 1\}$ be the binary indicator with $\alpha = \mathbb{P}(B = 0)$. Maximizing the mutual information $I(z; B_\alpha)$ encourages the learned representations to retain information about whether it originated from $\mathcal{X}$ or $\mathcal{S}$, supporting the goal of disentangling nuisance from task-relevant structure. The mutual information $I(z; B_\alpha)$ can be expressed in closed form. A standard result from information theory shows that, when $\alpha = 0.5$, it holds that $I(z; B_{0.5}) = \mathrm{JSD}(p(z \mid \mathcal{X}) \| p(z \mid \mathcal{S}))$. This also holds more generally, for any $\alpha$, when considering a family of weighted Jensen-Shannon divergences. Specifically:

$$I(z; B_\alpha) = \mathrm{JSD}_\alpha(p(z \mid \mathcal{X}) \| p(z \mid \mathcal{S})) = \alpha \, \mathrm{KL}(p(z \mid \mathcal{X}) \| M_\alpha) + (1 - \alpha) \, \mathrm{KL}(p(z \mid \mathcal{S}) \| M_\alpha), \tag{2}$$

where KL is the standard Kullback-Leibler divergence. This provides an estimator where the variance depends on the batch size instead of on the dimensionality of the feature space, and without any need for additional neural network parametrizations.

**Application to Colonoscopy**

To show the impact of leveraging side information on real-world applications, we evaluate our method on *Polyp histology classification*: classifying hyperplastic vs adenomatous polyps. We adapt the MSN framework [1] by incorporating our side information method. In addition to the original cross-entropy loss between anchor and target predictions $p^{(a)}$ and $p^{(t)}$ with ME-MAX regularization to avoid trivial solutions, we compute the JSD between aggregated anchor and target predictions across main and side samples.

$$\mathcal{L}_{\text{MSN-SI}} = \underbrace{\frac{1}{BM} \sum_{i=1}^{B} \sum_{j=1}^{M} H\left(p_i^{(t)}, p_{i,j}^{(a)}\right)}_{\text{cross-entropy}} - \lambda \underbrace{H\left(\bar{p}^{(a)}\right)}_{\texttt{ME-MAX}} - \gamma [\underbrace{\text{JSD}_\alpha\left(\bar{p}_{\mathcal{X}}^{(a)} \parallel \bar{p}_{\mathcal{S}}^{(t)}\right)}_{\text{anchor vs. side target}} + \underbrace{\text{JSD}\left(\bar{p}_{\mathcal{S}}^{(a)} \parallel \bar{p}_{\mathcal{X}}^{(t)}\right)}_{\text{side anchor vs. target}}]$$

**Colonoscopy Data.** For pre-training, we use REAL-Colon [4], a large and public dataset with around $2.7M$ frames from 60 recordings. REAL-Colon provides full length colonoscopy screenings, meaning that a majority of these frames are negatives without any polyps. There are in total $\sim 350K$ bounding box annotations, defining the set of positive images. The rest of the dataset is considered as the side information. For the downstream task we use PolypsSet [11], which provides bounding box annotations and binary labels for adenoma and hyperplastic polyps, with $\sim 38K$ frames from 155 video sequences split on sequence level into $75\%, 10\%, 15\%$ train, validation, and test. The learned representations are evaluated by linear probing. The results are compared to those reported by Hirsch et al. [9], noting that their models were pre-trained on different datasets—both public and private—than ours, which must be taken into account in the comparisons.

Table 1: F1 test performance on PolypsSet histology classification. Supervised learning (SL) and SSL pre-training on private and public datasets are compared. Note that data differs between our setting (bottom part) and that of Hirsch et al. [9] (upper part), their private data being one order of magnitude bigger than our public. This shows that our method learns useful features more efficiently.

| Method | Framework | Arch | Private | Public |
|---|---|---|---|---|
| FS [14] | SL | RN50 | - | 72.1 |
| DINO [14] | SSL | RN50 | - | 72.4 |
| MSN [9] | SSL | ViT-S | 78.5 | 70.6 |
| MSN [9] | SSL | ViT-B | 78.2 | 74.6 |
| MSN [9] | SSL | ViT-L | **80.4** | 73.6 |
| MSN | SSL | ViT-S | - | 76.1 |
| MSN-N (ours) | SSL | ViT-S | - | 77.8 |
| MSN-SI (ours) | SSL | ViT-S | - | **80.3** |

**Results.** We report macro-F1 test results for the polyp histology classification task on PolypsSet in Table 1. A model pre-trained on REAL-Colon with our choice for hyper-parameters (without incorporating side information) outperforms the best previous models pre-trained on public data by $1.5\%$, and by $5.5\%$ when comparing models with identical architectures, but underperforms when compared to models pre-trained on the larger private dataset. The naive incorporation MSN-N, where the model is trained with side information simply mixed into the dataset, improves the results by another $1.7\%$. However, when using our proposed method (MSN-SI), we achieve a F1-macro score of $80.3\%$, matching the best privately trained models *while using an order of magnitude less data and fewer parameters*. This demonstrates that, when informative data is limited but relevant side information is available, our method can learn useful features more efficiently — compensating for the data disadvantage through auxiliary structure.

**Potential Negative Societal Impact**

This work proposes a self-supervised learning framework for medical imaging that leverages side information — data assumed to contain task-irrelevant structures. While this approach has potential to improve model generalization and reduce reliance on large annotated datasets, its misuse or misinterpretation could have negative societal consequences. First, if the side information inadvertently includes task-relevant cues (e.g., anatomical or demographic markers), models trained under the Nuisance-Free MultiView assumption may learn biased or misleading representations, affecting diagnostic fairness across patient groups or imaging devices. Second, as with all automated systems in healthcare, deploying such models without sufficient clinical validation or human oversight could lead to diagnostic errors or misplaced trust in algorithmic outputs. These risks underscore the importance of responsible data management, rigorous evaluation across diverse populations and equipment, and transparent collaboration between AI researchers and clinicians prior to any clinical use.

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
