# OpenReview forum: "Improving Polyp Classification in Colonoscopy using Self-Supervised Learning with Side Information"
_EurIPS.cc/2025/Workshop/MedEurIPS — EurIPS 2025 Workshop MedEurIPS Submission_

### Official Review · Reviewer_StK5 · 2025-10-24
**Interesting idea with solid execution, minor issues to address**

**Rating:** 8
**Confidence:** 4

**Review:**

This paper presents a multi-view self-supervised learning framework that leverages side information to enforce invariance to task-irrelevant information shared across views. The work is well-motivated: the authors clearly articulate the importance of the downstream task (polyp classification) and justify their approach by noting that videos mostly contain redundant frames that are irrelevant to the task at hand. The proposed idea is novel and would be of interest to the workshop’s community.
However, some important implementation details are missing from the text:
- Based on the results in Table 1, the hyperparameters chosen during pre-training lead to substantially better results compared to the original MSN implementation trained on publicly available data, yet this is not thoroughly explained.
- It is unclear what “naive incorporation of side information” (MSN-N model) means in practice and how this affects the optimization objective.
- Limitations of this work are not discussed. For example, although the pre-training set is relatively small in size, it heavily relies on bounding box annotations to define positive images and side information, which might limit the scalability of the approach.

---

### Official Review · Reviewer_u322 · 2025-10-29
**Improving Polyp Classification in Colonoscopy using Self-Supervised Learning with Side Information**

**Rating:** 6
**Confidence:** 3

**Review:**

This work aimed to improve self-supervised learning for polyp classification in colonoscopy by incorporating side information. During training, the model was encouraged to separate relevant and irrelevant features, thus enhancing its robustness. The experimental results showed the competitiveness of this approach.

Pros:
- The method was novel in the context of polyp classification.
- The experiments illustrated the advantages of this method.

Cons:
- The structure of this paper could be improved, such as the headings and the lack of a conclusion.
- The experiments could be extended, like using a bigger network for the proposed method.

---

### Decision · Program_Chairs · 2025-10-31

**Decision:**

Accept (Poster)

**Comment:**

Both reviewers find the paper well motivated and relevant, proposing a novel use of side information in self-supervised learning for colonoscopy.